# Fungal Invasive Co-Infection Due to *Aspergillus fumigatus* and *Rhizopus arrhizus*: A Rhino-Orbital Presentation

**DOI:** 10.3390/jof7121096

**Published:** 2021-12-20

**Authors:** Juan Pablo Ramírez-Hinojosa, Salvador Medrano-Ahumada, Roberto Arenas, Arturo Bravo-Escobar, Sara Paraguirre-Martínez, Juan Xicohtencatl-Cortes, Erick Martínez-Herrera, Rigoberto Hernández-Castro

**Affiliations:** 1Departamento de Infectología, Hospital General Dr. Manuel Gea González, Tlalpan 14080, Mexico; dr.ramirezhinojosa@yahoo.com (J.P.R.-H.); dr.smedrano@gmail.com (S.M.-A.); 2Servicio de Micología, Hospital General Dr. Manuel Gea González, Tlalpan 14080, Mexico; rarenas98@hotmail.com; 3Servicio de Otorrinolaringología, Hospital General Dr. Manuel Gea González, Tlalpan 14080, Mexico; doctorbravo@gmail.com; 4Departamento de Patología, Hospital General Dr. Manuel Gea González, Tlalpan 14080, Mexico; parraguirres@hotmail.com; 5Laboratorio de Bacteriología Intestinal, Hospital Infantil de México Dr. Federico Gómez, Cuauhtémoc 06720, Mexico; juanxico@yahoo.com; 6Research Unit, Regional Hospital of High Specialty of Ixtapaluca, Ixtapaluca 56530, Mexico; 7Postgraduate Studies and Research Section, Higher School of Medicine, National Polytechnic Institute, Mexico City 11340, Mexico; 8Departamento de Ecología de Agentes Patógenos, Hospital General Dr. Manuel Gea González, Tlalpan 14080, Mexico

**Keywords:** mucormycosis, *Rhizopus arrhizus*, fungal infection, aspergillosis, *Aspergillus fumigatus*

## Abstract

Aspergillosis and mucormycosis are filamentous fungal infections occurring predominantly in immunocompromised patients. Fulminant process with rapid infiltration of the contiguous tissue is distinctive for both type of fungi. The rhinocerebral co-infection by *Aspergillus* and Mucorales is very rare and is usually associated in immunocompromised patients with a high mortality rate. This rare co-infection leads to difficulties in diagnosis, and therapeutic delays can result in a poor prognosis. Overall, the treatment of choice is surgical debridement and liposomal amphotericin B. This paper describes a combined aspergillosis and mucormycosis case in a diabetes mellitus type 2 patient with chronic ulcerations of the palatal and cheek. To our knowledge, this is the first report of an uncommon co-infection of *Aspergillus fumigatus* and *Rhizopus arrhizus* in a rhino-orbital presentation.

## 1. Introduction

Mucormycosis and invasive aspergillosis infections are aggressive fungal infections with a high mortality rate. These infections have generally predilection among patients with risk factors as immunosuppression, untreated diabetes mellitus, kidney diseases, hematological malignancies or mayor trauma [1,2,3]. Aspergillosis is the clinical syndrome caused by diverse species of *Aspergillus* genus; *A*. *fumigatus* is the most important pathogenic fungus in humans; however, other species had been associated in different pathologies. Some clinical presentations are often similar among *A*. *fumigatus* related species, and are frequently misidentified [4,5].

Following aspergillosis, mucormycosis is the second most frequently mycosis caused by filamentous fungi. Agents of mucormycosis belong to the order Mucorales. *Rhizopus*, *Mucor* and *Lichtheimia* are the main genera implicated as cause of mucormycosis [6]. The most common clinical form is the rhinocerebral mucormycosis, and *Rhizopus* is the etiological agent of 70% of the reported cases [2,7]. 

Reports of combined mucormycosis and aspergillosis infections limited to the oral-rhinocerebral region are exceptional. Diagnosis is often based on histopatological examination and supported by fungal isolation. The peculiarity of this infection leads to difficulties in diagnosis and delays can result in a poor prognosis [8,9]. This paper describes the clinical features of combined aspergillosis and mucormycosis case in a diabetes mellitus type 2 patient with a chronic intra-oral ulceration of the palatal and cheek.

## 2. Case

A 52-year-old male patient with a history of diabetes mellitus type 2 with apparently regular control was presented to the emergency room in January of 2016 with a 2-month history of swelling of the left side of the face, left facial palsy, ptosis, and an impaired vision. At clinical examination, there was prominent tumefaction of the left hemi-face, involving the peri-orbital, zygomatic and maxillary areas, naso-sinusal congestion and diffuse ulcerative and necrotic lesion of the palatal and cheek mucosa (Figure 1A,B). The peripheral blood tests revealed mild neutropenia (WBC 14,800/mL, with 87% neutrophils, and glucose 180 mg/dL, glycosylated haemoglobin 6.7%), no other relevant abnormalities were observed. During examination, a KOH smear from the necrotic ulcer was performed, which revealed hyaline branched coenocytic hyphae. Amphotericin B deoxycolate was initiated (1 mg/kg per day), and rhino-orbital surgical debridement was performed, as well as an orbital exenteration with partial maxillectomy. The pathology of the palate, maxillary sinus and ethmoid bone samples were consistent with mucormycosis, but some histological specimens proved an invasive growth with septate hyphae (Figure 2A,B). Based on the clinical findings and the morphological features of the fungal organisms, a presumptive diagnosis of mucormycosis and concomitant aspergillosis was formulated. Samples for mycological culture (nasal and maxillar-sinus) were cut into very thin pieces and homogenized before being cultivated on Sabouraud agar (SDA) for 10 days at room temperature. Nevertheless, fungal isolation was repeatedly negative. We decided to add caspofungin as a salvage therapy; the combined (amphotericin B + caspofungin) therapy was given for five days. Finally, the patient died after six weeks of treatment. 

This study was approved by the Institutional Review Board of Hospital General Dr. Manuel Gea Gonzalez, Mexico City, Mexico (24 August2018; IRB No. 06-77-2012-16).

Since the mycological culture was negative, it was decided to carry out the molecular identification in order to identify the causal agents associated with the patient. In this way, two paraffin-embedded tissue samples were analyzed, one sample from naso-sinus area and the second sample from maxillar sinus. The genomic DNA was isolated using a DNeasy blood and tissue kit (Qiagen, Ventura, CA, USA) according to the manufacturer’s instructions after preliminary removal of paraffin by extraction with xylene protocol. Molecular identification was performed by ITS rDNA region and beta-tubulin gene amplification and sequencing [10,11,12]. For ITS rDNA region amplification, a set of primers previously reported to identify fungi species (5′-TCCGTAGGTGAACCTTGCGG-3′) and ITS4 (5′-TCCTCCGCTTATTGATATGC-3′) were used [9,10]. The amplification program includes an initial denaturalization at 94 °C for 5 min; 33 cycles at 94 °C for 30 s, 55 °C for 35 s, 72 °C for 45 s, and a final extension for 5 min at 72 °C, with a no-template negative control. The reactions were performed in 50 µL reaction volume containing 25 µL of PCR master Mix (Qiagen, Ventura, CA, USA), 200 ng of DNA (2µL), 1 μL of each forward and reverse primer (25 pmol), and 21 μL nuclease-free water. 

For beta-tubulin gene amplification primers (5′-AATTGGTGCCGCTTTCTGG-3′) and (5′-AGTTGTCGGGACGGAATAG-3′) were used [12]. PCR reaction was performed in a final volume of 50 μL, containing of 2 μL of DNA (200 ng), 25 μL of PCR master mix, 1 μL of 50 pmol of forward and reverse primers, and 21 μL of nuclease-free water. The amplification program includes an initial denaturation at 94 °C for 5 min, followed by 33 cycles of denaturation at 94 °C for 30 s, annealing at 59.5 °C for 40 s, and extension at 72 °C for 45 s with a final extension at 72 °C for 10 min. The PCR products were separated on 1% agarose gels stained with ethidium bromide.

For ITS rDNA region amplification, a 517 bp product was amplified from naso-sinusal sample, and 620 bp product from maxillar sample. The PCR products were purified and the nucleotide sequence was determined in both directions using the same primers with Taq FS dye terminator cycle sequencing fluorescence-based sequencing and analyzed on an applied biosystems 3730 xl DNA sequencing system. The consensus sequences homology search was performed with ex-type ITS rRNA fungi databases (https://blast.ncbi.nlm.nih.gov/Blast.cgi, accessed on 9 December 2021). using BLASTn analysis and displayed 98.65% of homology with *A*. *fumigatus* ATCC 1022, *A*. *oerlinghausenensis* CBS139183 (98.85%), *A*. *fumisynnematus* IFM42277 (99.80%), *A*. *lentulus* NRRL 35552 (99.81%), *A*. *novofumigatus* CBS 117250 (99.61%), *A. fumigatiaffinis* CBS 117186 (99.42%), *A*. *fischeri* NRLL 181 (99.23%), *A*. *laciniosus* KACC 41657 (99.23%), *A*. *spinosus* NRRL 5034 (99.23%), *A*. *takakii* IFM 53599 (99.23%), *Aspergillus clavatus* ATCC 1007 95.42% and, *A. cervinus* NRRL 5025 (93.15%), among others species from naso-sinusal sample and 100% of homology with *Rhizopus arrhizus* (syn. *R. oryzae*) HP25 and 8–3 M strains, among others from maxillar samples [13].

The ITS rDNA region does not allow species-level discrimination for the genus *Aspergillus*, so molecular identification was performed using the beta-tubulin gene (*benA*). A 485 bp fragment from naso-sinusal and maxillar sample was amplified and the nucleotide sequence was determined in both directions using the same primers. The sequences homology search was performed with ex-type strains sequences under section *Aspergillus* subgen. *Fumigati* using BLASTn analysis The sequence showed a 98.74% of homology with *A*. *fumigatus* ATCC 1022, *A*. *oerlinghausenensis* CBS139183 (94.20%), *A*. *fischeri* NRLL 181 (93.08%), *A*. *spinosus* NRRL 5034 (91.84%), *A*. *laciniosus* KACC 41657 (91.18%), *A*. *takakii* IFM 53599 (91.08%), *A*. *lentulus* NRRL 35552 (90.59%), *A. fumigatiaffinis* CBS 117186 (90.38%), *A*. *novofumigatus* CBS 117250 (89.98%), *A*. *fumisynnematus* IFM42277 (89.25%), *Aspergillus clavatus* ATCC 1007 (78.10%) and, *A. cervinus* NRRL 5025 (77.57%) [13]. The sequences obtained from ITS rDNA region were assigned *Rhizopus arrhizus* (syn. *R*. *oryzae*) GEA-44, *Aspergillus fumigatus* GEA-44, and *Aspergillus fumigatus* GEA-1 for *benA* gene, sequences were deposited in the GenBank under accession number MG029627, MT340910 and MT370439, respectively.

## 3. Discussion

Aspergillosis and mucormycosis are filamentous fungal infections occurring predominantly in immunocompromised patients, and both types of infection are characterized for being fulminant processes with rapid infiltration of contiguous tissues. The most common cause of invasive aspergillosis is *A. fumigatus*, followed by *A. terreus*, *A. niger*, *A. flavus*, and other species belonging to *Fumigati* section [12]. *A. lentulus* has been described recently as a new species of *Aspergillus* genus; genetic analysis indicated its close relation with *A. fumigatus*, and it has also been reported as a cause of invasive aspergillosis [5,12]. The correct identification of *Aspergillus* is a challenge for both clinical microbiology and mycology laboratories, which complicates epidemiological investigations leading often to its misidentification as an atypical *A. fumigatus*. 

The classical fungi diagnosis involves a rapid microscopic identification using a KOH/chlorazol black preparation. The mycological culture is frequently reported as negative in half of the cases of mucormycosis. One of the main causes is the homogenization of the tissue since it can generate the loss of viability of the non-separated fragile hyphae of these fungi. In our clinical case, we failed at least three times in the mycological culture and one of the reasons was that we homogenized the tissue prior to culture. Therefore, it is recommended to slice and cut the tissue into very small and thin pieces before inoculation in the culture medium. Another problem associated with negative results is the use of a single culture medium (Sabouraud agar), so it is recommended to use other culture methods such as potato dextrose agar, Malt extract agar and Brain–heart–infusion (BHI) broth. The use of BHI broth provides optimal contact between the medium and the sample. In addition, growing at two temperatures (30 °C and 37 °C) can also increase yield of culture. Culture requires at least three days, and it needs mycology expertise for fungal description [14,15].

Histopathological examination displays characteristic fungal structures, such as large non-septate hyphae for Mucorales and septated branching hyphaes for *Aspergillus*; however, accurate and definitive identification is achieved by molecular methods [10,11,16]. Currently, one of the most used for fungi identification is the amplification and sequencing of ITS rRNA region, which allows the identification of a wide range of fungi. However, when this genetic marker cannot differentiate the species involved, it is necessary to use other genes such as beta-tubulin or Cadmoduline in the genus *Aspergillus*. For the molecular identification of *Aspergillus fumigatus*, experts recommend comparative sequence analysis of the ribosomal ITS region, specifically the ITS1 and ITS2 flanking regions of the 5.8S rDNA for the identification of the *Aspergillus Fumigati* section (intersection) and of the beta-tubulin or calmodulin genes for the identification of species level (intrasection). In this study, analysis of ITS rRNA region combined with amplification of *benA* gene, allowed the identification of the two etiologic fungal agents [5,12]. The correct identification of the causative agent is crucial to select the best systemic antifungal treatment which complements an aggressive surgical debridement of affected area. For both kinds of fungi, a fulminant process involving rapid infiltration of the neighboring tissues like the orbit and the anterior skull base is typical. A computed tomography (CT) scan is decisive to identify bone destruction, blood vessel compromise and spreading to adjacent structures (Figure 1C). MRI provides early detection of meningeal or intraparenchymatous spread as well as intracranial vascular occlusion. 

Both mucormycosis and aspergillosis are usually severe infections, with mortality rates ranging from 20% to 40% in immune-compromised patients and approaching to 80–90% in those cases of disseminated infections or involvement of the brain structures [2,7,8]. Appropriate treatment must be instituted early, even before the results of culture and/or histopathological studies have been obtained. Currently, surgical debridement of the lesions as well as systemic administration of amphotericin B are considered the standard therapy; posaconazole and isavuconazole are promising alternatives.

Table 1 shows the reported cases of combined aspergillosis and mucormycosis. The rhinocerebral form was the most frequent clinical presentation, but oro-sinonasal, orofacial, rhino-orbital as well as sinus and brain abscess were also reported. The most common underlying disease were diabetes mellitus, Castleman disease, acute myeloid leukemia, cerebral trunk glioma and renal transplantation. Most cases show amphotericin B as treatment of choice; however, voriconazole, posaconazole, itraconazole, fluconazole and micafungin have also been used. The etiological agent combinations were very diverse and poorly defined. 

Our case presented a rhino-orbital compromise, associated with diabetes mellitus, the most prevalent underlying disease, and was treated with amphotericin B, which is considered as a fundamental backbone of treatment for both agents; we also aimed to identify a mixed *A. fumigatus* and *R.*
*arrhizus* infection. Herein, to our knowledge, this is the first case of rhino-obital infection due to this rare fungal combination.

## Figures and Tables

**Figure 1 jof-07-01096-f001:**
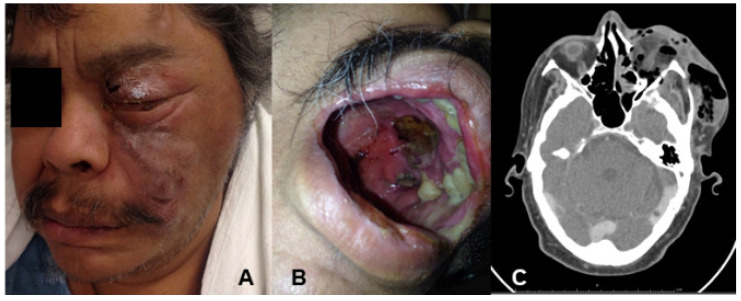
(**A**) Mucormycosis with facial infiltration and orbital involvement; (**B**) palatal necrotic lesions; (**C**) computed tomography with hypodense material involving multiple paranasal sinuses associated with bone erosion, and left orbit emphysema.

**Figure 2 jof-07-01096-f002:**
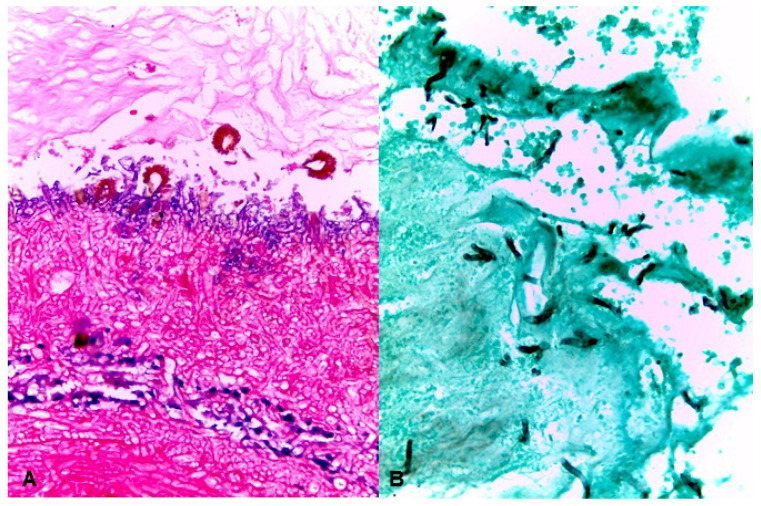
Nasal and maxillar-sinus tissue biopsies were paraffinembedded and formalin-fixed for histopathological examination. Three millimeter sections were stained with haematoxylin-eosin (H-e) and Gomori–Grocott methenamine-silver (G-G). (**A**) Nasal-sinus tissue sample showing the presnece of *Asperguillus* heads (H-E 40x); (**B**) Maxillar–sinus tissue sample showing presence of broad, non-septated hypahe (Gomori-Grocott 40x).

**Table 1 jof-07-01096-t001:** Combined aspergillosis and mucormycosis cases.

Case	Clinical Presentation	Underlying Disease	Treatment	Fungal Agents	Reference
1	Sinus	Chronic sinusitis	Surgery	*Aspergillus* spp./Mucorales	[17]
2	Oro-sinonasal	Castleman disease	Amphotericin B, and itraconazole	*Aspergillus* spp./Mucorales	[18]
3	Rhinocerebral	Diabetes mellitus	Voriconazole, caspofungin, and amphotericin B	*A*. *fumigatus*/Mucorales	[8]
4	Sinus and brain abscess	Diabetes mellitus	Amphotericin B	*A*. *fumigatus*/Mucorales	[19]
5	Sinonasal	Sinusitis	Itraconazole, and amphotericin B	*Aspergillosis*/Mucorales	[20]
6	Sinus	Acute myeloid leukemia	Posaconazole		[21]
7	Rhino-oculo-cerebral	Diabetes mellitus	Voriconazole	*A*. *flavus*/Mucorales	[22]
8	Rhinocerebral	Diabetes mellitus, and Renal transplantation	Amphotericin B	*A*. *niger*/*R*. *arrhizus* (syn. *R*. *oryzae*)	[23]
9	Orofacial	Cerebral trunk glioma	Fluconazole, amphotericin B, and micafungin	*A. flavus*/Mucorales	[3]
10	Rhinocerebral	Renal transplantation	Amphotericin B	*A. fumigatus*/ Mucorales	[24]
11	Rhino-orbital	Diabetes mellitus	Amphotericin B	*A. fumigatus*/*R*. *arrhizus* (syn. *R*. *oryzae*)	Present study

## Data Availability

The datasets used and/or analyzed during the present study are available from the corresponding author on reasonable request.

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
