# Peer review of "Fungal Invasive Co-Infection Due to Aspergillus fumigatus and Rhizopus arrhizus: A Rhino-Orbital Presentation"

_jof, 2021, doi:10.3390/jof7121096_

Round 1
Reviewer 1 Report
Dear Editor,
This is an interesting case report concerning a mixed fungal rhino-sinusitis with a rhino-orbital localisation.
The whole clinical presentation of the infection and its procedure is (unfortunately for the patients) quite typical, with only the exception of the co-infection by a mucormycete and an aspergillus. However, even this, is not an unexpected phenomenon in these patients.
Moreover, the diagnosis was not actually made by the molecular applications but, as it is already proposed, the latter served as a tool of genus-species identification and confirmation later of what was already visible in the tissue biopsies.
To my view, it is also peculiar that the cultures were constantly negative although there was a very heavy clinical image and extended lesions of that typical type. Were these cultures done in specific mycological media, were broth media used? Were all tissue parts sliced in very thin parts and inoculated in adequate depth? Probably it is not a bad idea to discuss this information in the text.
As for the molecular tools, there are not adequately described. Which were the molecular targets? They concerned an ITS or a 18S sequencing? Which were the primers? Which was the PCR used?
There are also some syntax or spelling mistakes (ex. analysis are not differentiate…was amplified and sequencing…micology laboratory…posaconazol…isavuconazol, etc.)
It is Rhizopus arrhizus nowadays and not Rhizopus oryzae.
The conclusion at the discussion section about the rhino-orbital infection due to a rare (not unique) fungal combination is more convincing than the one in the abstract about the molecular identification. Independently of the type of infection in these cases, we always have to confirm our phenotypic or histological identifications. It is a compulsory part of the mycological job.
Sincerely yours
Dear Editor,
This is an interesting case report concerning a mixed fungal rhino-sinusitis with a rhino-orbital localisation.
The whole clinical presentation of the infection and its procedure is (unfortunately for the patients) quite typical, with only the exception of the co-infection by a mucormycete and an aspergillus. However, even this, is not an unexpected phenomenon in these patients.
Moreover, the diagnosis was not actually made by the molecular applications but, as it is already proposed, the latter served as a tool of genus-species identification and confirmation later of what was already visible in the tissue biopsies.
To my view, it is also peculiar that the cultures were constantly negative although there was a very heavy clinical image and extended lesions of that typical type. Were these cultures done in specific mycological media, were broth media used? Were all tissue parts sliced in very thin parts and inoculated in adequate depth? Probably it is not a bad idea to discuss this information in the text.
As for the molecular tools, there are not adequately described. Which were the molecular targets? They concerned an ITS or a 18S sequencing? Which were the primers? Which was the PCR used?
There are also some syntax or spelling mistakes (ex. analysis are not differentiate…was amplified and sequencing…micology laboratory…posaconazol…isavuconazol, etc.)
It is Rhizopus arrhizus nowadays and not Rhizopus oryzae.
The conclusion at the discussion section about the rhino-orbital infection due to a rare (not unique) fungal combination is more convincing than the one in the abstract about the molecular identification. Independently of the type of infection in these cases, we always have to confirm our phenotypic or histological identifications. It is a compulsory part of the mycological job.
Sincerely yours
Author Response
Reviewer 1
Dear Editor,
This is an interesting case report concerning a mixed fungal rhino-sinusitis with a rhino-orbital localisation.
The whole clinical presentation of the infection and its procedure is (unfortunately for the patients) quite typical, with only the exception of the co-infection by a mucormycete and an aspergillus. However, even this, is not an unexpected phenomenon in these patients.
Moreover, the diagnosis was not actually made by the molecular applications but, as it is already proposed, the latter served as a tool of genus-species identification and confirmation later of what was already visible in the tissue biopsies.
To my view, it is also peculiar that the cultures were constantly negative although there was a very heavy clinical image and extended lesions of that typical type.
Were these cultures done in specific mycological media, were broth media used? Were all tissue parts sliced in very thin parts and inoculated in adequate depth? Probably it is not a bad idea to discuss this information in the text.
Answer: Two tissue samples from the most affected areas (naso and maxillar-sinus) were used, one for mycological culture and the other for histopathology (4 samples in total). The samples were taken by Infectologists and sent to the Microbiology and Pathology laboratory. The last sentence will be included in the mycological isolation section for better understanding.
Lines 72-75: Samples for mycological culture (nasal and maxillar-sinus) were cut into very thin pieces and homogenized before being cultivated on Sabouraud agar (SDA) for 10 days at room temperature.
As for the molecular tools, there are not adequately described. Which were the molecular targets? They concerned an ITS or a 18S sequencing? Which were the primers? Which was the PCR used?
Answer: This section was re-writing according to the two reviewers. The molecular targets were ITS rRNA region (the last nucleotides of the 18S subunit, ITS1-5.8S-ITS2 and the first nucleotides of the 28S) and beta-tubulin gene, both genes were sequencing. An end point PCR was used and primers and PCR protocols were included.
Lines 93-134: Since the mycological culture was negative, it was decided to carry out the molecular identification in order to identify the causal agents associated with the patient. In this way, two paraffin-embedded tissue samples were analyzed, one sample from naso-sinus area and the second sample from maxillar sinus. The genomic DNA was isolated using a DNeasy blood and tissue kit (Qiagen, Ventura CA, USA) according to the manufacturer’s instructions before preliminary removal of paraffin by extraction with xylene protocol. Molecular identification was performed by ITS rDNA region and beta-tubulin gene amplification and sequencing [9,10,11]. For ITS rDNA region amplification, a set of primers previously reported to identify fungi species (5'-TCCGTAGGTGAACCTTGCGG-3') and ITS4 (5'-TCCTCCGCTTATTGATATGC-3') were used [9,10]. The amplification program includes an initial denaturalization at 94°C for 5 min; 33 cycles at 94°C for 30 sec, 55°C for 35 sec, 72°C for 45 sec and a final extension for 5 min at 72°C, with a no-template negative control. The reactions were performed in 50 mL reaction volume containing 25 mL of PCR master Mix (Qiagen, Ventura, CA, USA), 200 ng of DNA (2 mL), 1 μL of each forward and reverse primer (25 pmol) and 21 μL nuclease-free water.
For beta-tubulin gene amplification primers (5'-AATTGGTGCCGCTTTCTGG-3 ') and (5'-AGTTGTCGGGACGGAATAG-3') were used [11]. PCR reaction was performed in a final volume of 50 μL, containing of 2 μL of DNA (200 ng), 25 μL of PCR master mix, 1 μL of 50 pmol of forward and reverse primers, and 21 μL of nuclease-free water. The amplification program includes an initial denaturation at 94°C for 5 min, followed by 33 cycles of denaturation at 94°C for 30 sec, annealing at 59.5°C for 40 sec, and extension at 72°C for 45 sec with a final extension at 72°C for 10 min. The PCR products were separated on 1% agarose gels stained with ethidium bromide.
For ITS rDNA region amplification, a 517 bp product was amplified from naso-sinusal sample, and 620 bp product from maxillar sample. The PCR products were purified and the nucleotide sequence was determined in both directions using the same primers with Taq FS dye terminator cycle sequencing fluorescence-based sequencing and analyzed on an applied biosystems 3730 xl DNA sequencing system. The consensus sequences homology search was performed using BLASTn analysis and displayed 99% of homology with Aspergillus lentulus IFM61591, A. fumigatus S30, and A. fumisynnematus IFM42277, among others species from naso-sinusal sample and 100% of homology with Rhizopus arrhizus (syn. R. oryzae) HP25 and 8–3 M strains, among others from maxillar sample.
The ITS rDNA region do not allow species-level discrimination for the genus Aspergillus, so molecular identification was performed using the beta-tubulin gene (benA). A 485 bp fragment from naso-sinusal and maxillar sample was amplified and the nucleotide sequence was determined in both directions using the same primers. The sequence showed a 100% of homology with A. fumigatus FH99 strains, and 99% of homology with A. fumigatus ATCC204305, and CYH5 among others strains. The sequence obtained from ITS rDNA region was called Rhizopus arrhizus (syn. R. oryzae) GEA-44 and Aspergillus fumigatus GEA-1 for benA gene, both sequences were deposited in the GenBank under accession number MG029627and MT370439, respectively.
There are also some syntax or spelling mistakes (ex. analysis are not differentiate…was amplified and sequencing…micology laboratory…posaconazol…isavuconazol, etc.)
Answer: Syntax and spelling mistakes were changed
It is Rhizopus arrhizus nowadays and not Rhizopus oryzae.
Answer: Rhizopus oryzae was changed for Rhizopus arrhizus in all manuscript, including table 1.
The conclusion at the discussion section about the rhino-orbital infection due to a rare (not unique) fungal combination is more convincing than the one in the abstract about the molecular identification. Independently of the type of infection in these cases, we always have to confirm our phenotypic or histological identifications. It is a compulsory part of the mycological job.
Answer: We modify the conclusion in the abstract.
Lines 31-32: To our knowledge this is the first report of an uncommon co-infection of Aspergillus fumigatus and Rhizopus arrhizus in a rhino-orbital presentation.

Reviewer 2 Report
Dear authors,
thank you very much for the preparation of the manuscript. I welcome case reports providing accurate species identifications because they are very valuable for the clinical mycology community in understanding epidemiology, and I hope this will become standard in case reports.
However, your manuscript needs revision in the following points:
- Taxonomy: The taxonomy of clinical recently fungi is a fast-changing field. Concerning the Mucorales, I like to recommend this review: Walther, G.; Wagner, L.; Kurzai, O. Updates on the taxonomy of Mucorales with an emphasis on clinically important taxa. Journal of Fungi 2019, 5, doi:10.3390/jof5040106.
- The methology of the molecular diagnosis needs to be reworked (see below) and the sequences should be made available on GenBank.
- The discussion needs to be carefully revised with regard to language and wording.
Some aspects in detail:
- Italic notation for section, genus, and epithet.
- Check typeset: hyphen-breaks especially in the introduction
- Line 27: All Mucormyosis-causing taxa belong to the order Mucorales. The term zygomycetes is now only informally used and should be avoided since it represents a polyphyletic group (Spatafora, J.W.; Chang, Y.; Benny, G.L.; Lazarus, K.; Smith, M.E.; Berbee, M.L.; Bonito, G.; Corradi, N.; Grigoriev, I.; Gryganskyi, A., et al. A phylum-level phylogenetic classification of zygomycete fungi based on genome-scale data. Mycologia 2016, 108, 1028-1046, doi:10.3852/16-042.)
- Line 32: The offical name of Rhizopus oryzae is Rhizopus arrhizus (http://www.speciesfungorum.org). It can be referred as “Rhizopus arrhizus (syn. Rhizopus oryzae)”.
- Line 42: fungus
- Line 45: That’s not correct! (1126/scitranslmed.3004404 ) Following aspergillosis, mucormycosis is the second most frequent mycosis caused by filamentous fungi. Agents of mucormycosis belong to the order Mucorales…
- Line 46: Absidia: the mucormycosis causing taxa of Absidia are now classified in the genus Lichtheimia (see e.g., https://doi.org/10.1128/AAC.01270-09 and https://doi.org/10.3390/jof5040106).
- Line 52-53: What are these difficulties in diagnosis? Readers might not be curious.
- Line 73-75: Please reconsider this sentence. There seems to be something wrong.
- Line 77: “The patient died after 6 weeks of in hospital treatment”
- To figure 2: Please add some more details concerning the sample and sampling methods for readers that are not familiar with the diagnostic standards.
- Line 87-108: These paragraphs need rework. Here are some thoughts:
- 18S-ITS1-5.8S-ITS2-28S rRNA can also be referred as ITS rDNA region
- The method order is confusing. Consider to follow this order: DNA extraction, DNA-amplification, sequencing, BLAST analyses
- Please consider to make the sequences available on GenBank. This will help to evaluate and value your finding in the future with regard to molecular classification.
- BLASTn analysis: Please restrict you blastn to “Sequence from type material” and mention also the GenBank accession number of the corresponding hit.
- Line 104-105: Alternative proposal: “The 18S-ITS1-5.8S-ITS2-28S rRNA do not allow species-level discrimination for the genus Aspergillus, so molecular identification was performed using the beta-tubulin gene (benA).”
- Table 1. Please be careful with using Mucor as generalization for agents of mucormycosis! In the case reports that you cite, the agents of mucormycosis are often not even identified to the genus level. I recommend using the term “Mucorales” in the column “Fungal agents” (also for “Zygomycetes” and “Mucormycosis”) unless otherwise stated in the corresponding case report. In fact, “Mucor” is often used in a misleading and wrong way, as in your reference 18.
Author Response
Reviewer 2
Dear authors,
thank you very much for the preparation of the manuscript. I welcome case reports providing accurate species identifications because they are very valuable for the clinical mycology community in understanding epidemiology, and I hope this will become standard in case reports.
However, your manuscript needs revision in the following points:
- Taxonomy: The taxonomy of clinical recently fungi is a fast-changing field. Concerning the Mucorales, I like to recommend this review: Walther, G.; Wagner, L.; Kurzai, O. Updates on the taxonomy of Mucorales with an emphasis on clinically important taxa. Journal of Fungi 2019, 5, doi:10.3390/jof5040106.
Answer: We agree with the reviewer and have used the term Mucorales throughout the manuscript, including table 1. The reference, Walther et al., 2019 was included.
- The methology of the molecular diagnosis needs to be reworked (see below) and the sequences should be made available on GenBank.
Answer: Lines 93-134: This section was rewritten and the sequences were submitted to GenBank
- The discussion needs to be carefully revised with regard to language and wording.
Answer: The discussion was carefully revised
Some aspects in detail:
- Italic notation for section, genus, and epithet.
Answer: The changes were made to italics (Introduction section)
- Check typeset: hyphen-breaks especially in the introduction
Answer: The hyphen-breaks were placed by the journal format
- Line 27: All Mucormyosis-causing taxa belong to the order Mucorales. The term zygomycetes is now only informally used and should be avoided since it represents a polyphyletic group (Spatafora, J.W.; Chang, Y.; Benny, G.L.; Lazarus, K.; Smith, M.E.; Berbee, M.L.; Bonito, G.; Corradi, N.; Grigoriev, I.; Gryganskyi, A., et al. A phylum-level phylogenetic classification of zygomycete fungi based on genome-scale data. Mycologia 2016, 108, 1028-1046, doi:10.3852/16-042.).
Answer: We agree and eliminate zygomycetes by Mucorales
Line 32: The offical name of Rhizopus oryzae is Rhizopus arrhizus (http://www.speciesfungorum.org). It can be referred as “Rhizopus arrhizus (syn. Rhizopus oryzae)”.
Answer: We agree to the official name and change them throughout the manuscript.
- Line 42: fungus
Answer: Fungi was changed for fungus
- Line 45: That’s not correct! (1126/scitranslmed.3004404) Following aspergillosis, mucormycosis is the second most frequent mycosis caused by filamentous fungi. Agents of mucormycosis belong to the order Mucorales…
Answer: The change was made as suggested by the reviewer.
Lines 45-47: Following aspergillosis, mucormycosis is the second most frequently mycosis caused by filamentous fungi. Agents of mucormycosis belong to the order Mucorales
- Line 46: Absidia: the mucormycosis causing taxa of Absidiaare now classified in the genus Lichtheimia (see e.g., https://doi.org/10.1128/AAC.01270-09 and https://doi.org/10.3390/jof5040106).
Walther, G.; Wagner, L.; Kurzai, O. Updates on the Taxonomy of Mucorales with an Emphasis on Clinically Important Taxa. J Fungi (Basel) 2019, 5, 106.
- Line 52-53: What are these difficulties in diagnosis? Readers might not be curious.
Answer: Difficulties in diagnosis are usually associated with the techniques used routinely. Histopathology which only allows us to observe the characteristics of the hyphae (mycelium) and the isolation of the fungus, where the success of the isolation is not always certain. In addition to the time spent in isolation, phenotypic characteristics do not always allow the final identification of the causative agent. Molecular identification allows us to define the genus and species with greater precision and specificity. However, in most Latin American countries, molecular identification is difficult to perform, as well as in many countries in other parts of the world.
- Line 73-75: Please reconsider this sentence. There seems to be something wrong.
Answer: We decided to delete this sentence.
Lines 75-78: “Despite an aggressive pharmacologic, surgical and antifungal treatment with amphotericin B during 41 days, persistent purulent drainage from oral and ophthalmic cavity and posaconazole was unavailable”.
- Line 77: “The patient died after 6 weeks of inhospital treatment”
Answer: Line 80: in hospital was eliminated of the sentence.
- To figure 2: Please add some more details concerning the sample and sampling methods for readers that are not familiar with the diagnostic standards.
Answer: Lines 87-91: The figure legend was modified
Figure 2. Nasal and maxillar-sinus tissue biopsies were paraffinembedded and formalin-fixed for histopathological examination. A 3 mm sections were stained with haematoxylin-eosin (H-e) and Gomori-Grocott methenamine-silver (G-G). (a) Nasal-sinus tissue sample showing the presnece of Asperguillus heads (H-E 40x). (b) Maxillar–sinus tissue sample showing presence of broad, non-septated hypahe (Gomori-Grocott 40x).
- Line 87-108: These paragraphs need rework. Here are some thoughts:
- 18S-ITS1-5.8S-ITS2-28S rRNA can also be referred as ITS rDNA region
- The method order is confusing. Consider to follow this order: DNA extraction, DNA-amplification, sequencing, BLAST analyses
- Please consider to make the sequences available on GenBank. This will help to evaluate and value your finding in the future with regard to molecular classification.
- BLASTn analysis: Please restrict you blastn to “Sequence from type material” and mention also the GenBank accession number of the corresponding hit.
Answer: Lines 93-134: This section was rewritten taking into consideration the suggestions of the reviewers.
- Line 104-105: Alternative proposal: “The 18S-ITS1-5.8S-ITS2-28S rRNA do not allow species-level discrimination for the genus Aspergillus, so molecular identification was performed using the beta-tubulin gene (benA).”
Answer: The change was made according to the reviewer
Table 1. Please be careful with using Mucor as generalization for agents of mucormycosis! In the case reports that you cite, the agents of mucormycosis are often not even identified to the genus level. I recommend using the term “Mucorales” in the column “Fungal agents” (also for “Zygomycetes” and “Mucormycosis”) unless otherwise stated in the corresponding case report. In fact, “Mucor” is often used in a misleading and wrong way, as in your reference 18.
Answer: We agree; in most cases the identification is at the genus level. So we make the changes suggested by the reviewer.

Round 2
Reviewer 1 Report
Dear Editor,
I still feel that the necessary cultures were not done correctly and although the confirmation of the diagnosis was done molecularly this could have been already revealed by a meticulous classical investigation, meaning direct microscopy and culturing.
The homogenisation of the biopsies (lines 86-89) before culturing for fungi is a crucial mistake and especially when investigating for mucorales. The procedure should be based on slicing and cutting in very small and thin pieces before inoculation but never homogenisation due to the destruction of the hyphae. Also several culture media should be used as well as broth media.
I disagree also with the comment in line 160 and below. Although a multi locus analysis is a very good option, ITS rDNA region is a very good locus for species-level discrimination for the genus Aspergillus as well.
Sincerely yours.
Author Response
Dear Editor,
I still feel that the necessary cultures were not done correctly and although the confirmation of the diagnosis was done molecularly this could have been already revealed by a meticulous classical investigation, meaning direct microscopy and culturing.
The homogenisation of the biopsies (lines 86-89) before culturing for fungi is a crucial mistake and especially when investigating for mucorales. The procedure should be based on slicing and cutting in very small and thin pieces before inoculation but never homogenisation due to the destruction of the hyphae. Also several culture media should be used as well as broth media.
Answer: We agree with the comment, the homogenization of the biopsies is a big mistake when performing the mycological culture; However, the diagnosis was made by the Hospital's diagnosis section (Microbiology), the molecular identification that we carried out allowed us to identify the two fungi involved. We propose to include this sentence in the discussion to mark the error and propose the recommendations of the references that are included.
Page 4; Line 144-155: The mycological culture is frequently reported as negative in half of the cases of mucormycosis. One of the main causes is the homogenization of the tissue since it can generate the loss of viability of the non-separated fragile hyphae of these fungi. In our clinical case, we failed at least three times in the mycological culture and one of the reasons was that we homogenized the tissue prior to culture. Therefore, it is recommended to slice and cut the tissue into very small and thin pieces before inoculation in the culture medium.
Another problem associated with negative results is the use of a single culture medium (Sabouraud agar), so it is recommended to use other culture methods such as potato dextrose agar, Malt extract agar and Brain-heart-infusion (BHI) broth. The use of BHI broth provides optimal contact between the medium and the sample. In addition, growing at two temperatures (30°C and 37°C) can also increase yield of culture.
- Cornely, O. A., Alastruey-Izquierdo, A., Arenz, D., Chen, S., Dannaoui, E., Hochhegger, B., Hoenigl, M., Jensen, H. E., Lagrou, K., Lewis, R. E., Mellinghoff, S. C., Mer, M., Pana, Z. D., Seidel, D., Sheppard, D. C., Wahba, R., Akova, M., Alanio, A., Al-Hatmi, A., Arikan-Akdagli, S., … Mucormycosis ECMM MSG Global Guideline Writing Group (2019). Global guideline for the diagnosis and management of mucormycosis: an initiative of the European Confederation of Medical Mycology in cooperation with the Mycoses Study Group Education and Research Consortium. The Lancet. Infectious diseases, 19 (12), e405–e421. https://doi.org/10.1016/S1473-3099(19)30312-3
- Lass-Flörl C. (2009). Zygomycosis: conventional laboratory diagnosis. Clinical microbiology and infection.15 Suppl 5, 60–65. https://doi.org/10.1111/j.1469-0691.2009.02999.x
I disagree also with the comment in line 160 and below. Although a multi locus analysis is a very good option, ITS rDNA region is a very good locus for species-level discrimination for the genus Aspergillus as well.
Answer: We understand your comment; however, numerous publications emphasize that the ITS rRNA region cannot differentiate the Aspergillus Fumigati section at the species level, that was the reason why it was decided to use the beta-tubulin gene. We propose to use the following sentence to give the utility of ITS rRNA region in Aspergillus Fumigati section.
We also include references where it is specified that the ITS rRNA region can identify at the section level (Aspergillus Fumigati section).
Page 5; Lines 164-168: For the molecular identification of Aspergillus fumigatus, experts recommend comparative sequence analysis of the ribosomal ITS region, specifically the ITS1 and ITS2 flanking regions of the 5.8S rDNA for the identification of the Aspergillus Fumigati section (intersection) and of the beta-tubulin or calmodulin genes for the identification of species level (intrasection).
Lamoth F. (2016). Aspergillus fumigatus-Related Species in Clinical Practice. Frontiers in microbiology, 7, 683. https://doi.org/10.3389/fmicb.2016.00683
Balajee, S. A., Houbraken, J., Verweij, P. E., Hong, S. B., Yaghuchi, T., Varga, J., et al. (2007). Aspergillus species identification in the clinical setting. Stud. Mycol. 59, 39–46. doi: 10.3114/sim.2007.59.05
Balajee, S. A., Borman, A. M., Brandt, M. E., Cano, J., Cuenca-Estrella, M., Dannaoui, E., et al. (2009a). Sequence-based identification of Aspergillus, fusarium, and mucorales species in the clinical mycology laboratory: where are we and where should we go from here? J. Clin. Microbiol. 47, 877–884. doi: 10.1128/JCM.01685-08
Hong, S. B., Go, S. J., Shin, H. D., Frisvad, J. C., and Samson, R. A. (2005). Polyphasic taxonomy of Aspergillus fumigatus and related species. Mycologia 97, 1316–1329. doi: 10.3852/mycologia.97.6.1316
Yaguchi, T., Horie, Y., Tanaka, R., Matsuzawa, T., Ito, J., and Nishimura, K. (2007). Molecular phylogenetics of multiple genes on Aspergillus section Fumigati isolated from clinical specimens in Japan. Nippon Ishinkin Gakkai Zasshi 48, 37–46. doi: 10.3314/jjmm.48.37
Samson, R. A., Hong, S., Peterson, S. W., Frisvad, J. C., and Varga, J. (2007). Polyphasic taxonomy of Aspergillus section Fumigati and its teleomorph Neosartorya. Stud. Mycol. 59, 147–203. doi: 10.3114/sim.2007.59.14
Alcazar-Fuoli, L., Mellado, E., Alastruey-Izquierdo, A., Cuenca-Estrella, M., & Rodriguez-Tudela, J. L. (2008). Aspergillus section Fumigati: antifungal susceptibility patterns and sequence-based identification. Antimicrobial agents and chemotherapy, 52(4), 1244–1251. https://doi.org/10.1128/AAC.00942-07
Samson, R. A., Visagie, C. M., Houbraken, J., Hong, S. B., Hubka, V., Klaassen, C. H., et al. (2014). Phylogeny, identification and nomenclature of the genus Aspergillus. Stud. Mycol. 78, 141–173. doi: 10.1016/j.simyco.2014.07.004
Simões D, Aranha Caetano L, Veríssimo C, Viegas C, Sabino R. Aspergillus collected in specific indoor settings: their molecular identification and susceptibility pattern. Int J Environ Health Res. 2021;31(3):248-257
Sincerely yours.

Reviewer 2 Report
Dear authors,
the report is much improved and there are only small things that need to be corrected. Thank you for making the sequences available!
Line 67: “de-scribes” still needs to be corrected
Line 119: before à after
page 4 first paragraph: Text indent was slightly shifted to the right
Line 166: called à assigned to
Line 172: Is the comma intended?
Line 183: Mucorales (instead of Mucor)
Line 188: Calmodulin
Line 188/189: “especially for the identification of Aspergillus genus.” Alternative: “e.g., in the genus Aspergillus.”
Line 416 and 417: commas are missing before “and” in both lists.
Table1: Case 9: Please erase “(syn. R. oryzae)”.
Author Response
Review II
the report is much improved and there are only small things that need to be corrected. Thank you for making the sequences available!
Line 67: “de-scribes” still needs to be corrected
Answer: The word was corrected
Line 119: before à after
Answer: The word was changed
page 4 first paragraph: Text indent was slightly shifted to the right
Answer: The format of the journal is modified when making the changes, we will send a message pointing out these changes to the editor.
Line 166: called à assigned to
Answer: The word was changed
Line 172: Is the comma intended?
Answer: The comma was eliminated
Line 183: Mucorales (instead of Mucor)
Answer: The word was changed
Line 188: Calmodulin
Answer: The word was changed
Line 188/189: “especially for the identification of Aspergillus genus.” Alternative: “e.g., in the genus Aspergillus.”
Answer: The sentence was changed
Line 416 and 417: commas are missing before “and” in both lists.
Answer: comas were included
Table1: Case 9: Please erase “(syn. R. oryzae)”.
Answer: (syn. R. oryzae) was eliminated
